# In Vivo Complete-Arch Implant Digital Impressions: Comparison of the Precision of Three Optical Impression Systems

**DOI:** 10.3390/ijerph19074300

**Published:** 2022-04-03

**Authors:** Jaime Orejas-Perez, Beatriz Gimenez-Gonzalez, Ignacio Ortiz-Collado, Israel J. Thuissard, Andrea Santamaria-Laorden

**Affiliations:** 1Faculty of Biomedical and Health Sciences, Department of Clinical Dentistry, Universidad Europea de Madrid, 28670 Madrid, Spain; jaime.orejas@universidadeuropea.es (J.O.-P.); ignacio.ortiz@universidadeuropea.es (I.O.-C.); 2Department of Implantology and Prosthetic Dentistry, Academic Center for Dentistry Amsterdam ACTA, 1081 LA Amsterdam, The Netherlands; b.gimenezgonzalez@acta.nl; 3Faculty of Biomedical and Health Sciences, Department of Medicine, Universidad Europea de Madrid, 28670 Madrid, Spain; israeljohn.thuissard@universidadeuropea.es

**Keywords:** optical impression, intraoral scanners, edentulous, complete arch, dental implants, precision

## Abstract

(1) Multiple in vitro studies reported insufficient accuracy of intraoral scanners (IOSs) for complete-arch multiple implant impression. The aim of the study is to analyze the precision of three IOSs, PIC dental (Pic dental, Iditec North West SL), TRIOS 3 (3Shape), and True Definition (Midmark Corporation) and the influence of several factors in the edentulous complete maxillary and mandibular arch. (2) A fully edentulous patient with eight implants in the maxillary and in the mandibular jaw was selected. Five impressions were taken per system and arch. A suprastructure was designed on each digital working cast. The precision was analyzed comparing each of the 28 distances and seven relative angulations of the abutments of all the designed suprastructures. The descriptive statistics, the Student’s *t*-test, and the ANOVA test were used to analyze the data (α = 0.05). (3) Significant differences were observed when comparing the IOSs in some of the distances and angulations. (4) The increase in the distance between implants affected the precision of T and TD but not the PIC system. The type of arch did not affect the PIC precision, but the T and TD systems performed worse in the mandibular arch. The system with the best precision was the PIC, followed by TD, and then T.

## 1. Introduction

The majority of studies that evaluate IOSs are performed in vitro and, therefore, lack the challenges that the IOSs face in vivo [1,2,3,4,5]. Despite having less clinical relevance, in vitro studies help to develop protocols and to analyze parameters that are not possible in vivo, such as accuracy [6,7,8]. The heterogeneity of these study designs makes the interpretation of the conclusions and clinical recommendations difficult [1,9,10,11].

The different IOSs in the market compete to achieve the best accuracy, defined as how closely the obtained measurements resemble the real arch measurements, and the best precision, defined as how similar the obtained measurements of the repeated scans are. (ISO 5725) [12]. Evaluating the precision in vivo is key, because there are many factors that can affect the result in the clinical setting, making the outcomes inconsistent and therefore difficult to predict. A large challenge is that there is no consensus regarding the range of acceptable misfit and the way to correctly measure the misfit clinically [9,13,14,15,16,17]. When the number of implants in the same structure increases, the tolerance of the error in the axis (X, Y, Z) and the angulations decrease [18]. Furthermore, we must consider the manufacturing tolerance of the suprastructures that can generate misfits in form of GAPs between 20 and 100 microns [19,20].

The majority of the literature agrees, that in order to perform complete arch implant impressions, the conventional techniques are superior to the digital ones [3,21,22,23,24,25,26]. The hardware and software technology used by the IOSs to capture images, the scan protocol used, the algorithms needed to create the STL file, and the transformation of the STL on the 3D working cast affect the definitive result [2,27,28,29,30,31,32,33]. Other factors such as the mouth opening of the patient, the size of the tip, the ambient light, light reflection, saliva, steam, the number and 3D position of the scan bodies, the manufacturing material of the scan bodies, the distance between them, and the length of the edentulous span and arch length are also described as factors that affect this technology [2,34,35,36,37].

Despite several IOSs not including complete arch implant impressions in their official indications, many articles evaluate that possibility. There are several factors affecting the accuracy and precision of the IOSs, long edentulous spans (lacking references), mobile tissues, number of implants, the stitching of the 3D images to produce the STL file, which is more challenging with the anatomy of the scan bodies and edentulous spans [14,27,38,39,40]. This is true for all the IOSs in the market in different measures, two of which are analyzed in this study, Trios (T) and True Definition (TD) [41,42]. However, the third scanner used in this study was PIC Dental (PIC), which was specifically designed for complete arch multiple implant impressions, adapting the stereophotogrammetry technology to the oral cavity [43,44,45,46,47,48].

More in vivo studies are needed in order to analyze these systems in real clinical situations [1]. Comparing these results with those of the in vitro studies will help understand the impact of the oral environment and the real idea of the possible indications of each IOSs [1].

The aims of this in vivo study are: (1) to analyze the precision of the three IOSs and compare them using arbitrary limits of 75 microns per span and 0.6 degrees of angulation for each scan body, (2) to evaluate the effect of the increasing distance between the implants, from the first scanned implant to the eighth implant measuring the Euclidean distances and relative angulations, and (3) to evaluate the effect of the arch (maxillary or mandibular) in the precision of the different scanners. Therefore, the corresponding null hypotheses are: H01 There are no differences in precision between the three IOSs using the established limits as a reference. H02 The increase in distance does not affect the precision of the three IOSs. H03 The precision of the IOSs is not affected by the type of arch (maxillary or mandibular).

## 2. Material and Methods

Overview: Three digital impression systems were evaluated in the present study; PIC dental (PIC), Trios (T), and True Definition (TD). For this, 5 impressions were taken per arch and system in one patient with 8 implants in each arch. Every STL was transformed in its corresponding digital working cast (Figure 1). A total of 30 digital suprastructures were designed. The Euclidean distances and angulations of the scan bodies were the analyzed variables. The measurements were performed in the design of the suprastructures [14,49,50] (Figure 2).

For the study, a 65-year-old male patient classified as ASA 2, fully edentulous with 8 external hex implants with platform 4,1 and 5 mm (Medical Precision Implants MPI, Madrid, Spain) in each arch was selected. The Regional Medical Ethical committee approved the study on 7 October 2014, with the code TESISIMPLAN2014, with version 2.0, 26 September 2014. Subsequently, the Research Committee of the European University of Madrid (UEM) issued the approval on 17 November 2014 with the code CIPE/009/14.

The null hypotheses of the study were: H01 There are no differences in precision between the 3 IOSs using the established limits as a reference. H02 The increase in distance does not affect the precision of the 3 IOSs. H03 The precision of the IOSs is not affected by the type of arch (maxillary or mandibular). Assuming a 20% probability of improvement in accuracy (alternative hypothesis), it was calculated that 5 repetitions for each impression, techniquem and arch on one patient were necessary to detect the clinically minimal relevant effect (OR = 1.5) with a confidence level of 95% and a statistical power of 80%.

### 2.1. Impression Systems and Impression Techniques

The impression taking was randomized using the statistical epidemiological program (EPIDAT 3.1). Each day an impression was taken per system and arch. The impressions with the systems T and TD were taken with the same scan bodies placing them only once to avoid the effect of that step. The first impression was always taken with T because it did not need a powder spray.

### 2.2. PIC Dental

Impressions: Position Implants Correctly dental (Iditec North West SL) The system was created specifically to capture multiple implants intraorally. The technology is based on stereophotogrammetry. The PIC Camera registers the implants placed in the oral cavity with its software (PIC pro) (Pic dental, Iditec North West SL) and the specific and distinct identification of the scan flags named PIC Transfers. This produces a file (PIC File) in real time with the relative distances and angulations of the implants. The PIC Transfers were manufactured for this study and placed by the same operator with manual torque, making sure that there was correct visibility for the camera lenses [51].

### 2.3. Trios 3 Pod 3Shape (T)

Impressions: Trios (3Shape A/S Copenhagen, Denmark) The system uses the technology named parallel confocal microscopy [8]. Eight scan bodies were used “Elos Accurate 6A-B Scan Body” Brånemark System RP (REF IO 6A-B; LOT 125149), along with the corresponding screwdriver “Elos Accurate IO Driver Short” (REF C13485) ELOS MEDTECH dental (Pinol A/S, Gørløse, Denmark) with 10 Nm [7].

### 2.4. True Definition (TD)

Impressions: Midmark (Midmark Corporation, Dayton, OH, USA). The technology used is called active wavefront sampling (AWS) and uses a titanium dioxide powder coating. The same scan bodies as with T were used, (ELOS) [7].

The impressions were taken at the University Clinic of the UEM by the same operator with more than 3 years of experience with the systems. The scanning protocols followed the instructions of the manufacturer starting always in implant 46 for the mandibular arch and 16 for the maxillary arch for T and TD. Some modifications were needed in the scanning with T and TD in order to be able to finish the impressions due to errors in the stitching of the scan bodies.

### 2.5. Digital Working Casts

The impressions obtained by T and TD were transformed into the digital working cast by aligning with the best fit algorithm the scan bodies with the equivalent CAD design from the implant library of the EXOCAD (EXOCAD 2.0.0.0 GmbH, Darmstad, Germany) This step was not necessary with the PIC system, as described previously (Figure 1).

With the aim to reach the definitive step of the CAD workflow, a suprastructure was designed for each digital working cast with Exocad [52]. Each suprastructure contained 8 rotatory abutments and 7 Ackerman segments with 3 mm sections. There was no virtual space between the suprastructure and the implant; hence, the position and the results are not affected. Both steps were performed by the same operator.

### 2.6. Assessment of the Distances and Angulations

The designed suprastructure (*n* = 30) was exported in STL format to a reverse engineering software, Geomagic (Geomagic Inc., 3D Systems, Morrisville, NC, USA)) [52]. The central points of each abutment platform were calculated (*n* = 8). The vector that passes through those points following the rotational axis of each abutment was calculated as well. The Euclidean distances were calculated connecting the central points between the segments, creating all possible combinations per suprastructure (*n* = 28). The relative angulations were calculated in degrees using the vectors of each abutment. The axis (0, 0, 0) was selected in the abutments corresponding to implant 46 in the mandibular arch and 16 in the maxillary arch. That same abutment was determined as (0, 0, 1) to calculate the angulations; in this way, abutments 46 and 26 were the reference to calculate the 7 remaining angulations (Figure 2). The Euclidean distances and angulations were calculated individually (one by one).

Due to the lack of consensus regarding the tolerable limits of misfit between the abutments and their angulations, and in order to help interpret the data, a limit of 75 microns per segment and 0,6 degrees, based on the available literature, were used to compare the precision [53,54] (Figure 3).

### 2.7. Statistical Analysis

A descriptive analysis of the data was carried out using the mean, standard deviation (SD), and maximum and minimum deviation of the distances and angulations of the abutments after the parametric behavior of the variables using the Shapiro–Wilk test was verified. The existence of statistically significant differences in the distances and angulations for the different intraoral scanners used was analyzed by means of the parametric Student’s t- and ANOVA tests, in which the Bonferroni adjustment techniques were applied for multiple comparisons.

For each system used, the variability of the measurements for the different distances and angulations was analyzed using Levene’s test of homogeneity of variances.

All the analyses were performed two-tailed, considering a significant *p*-value when it was below the alpha error (*p* < 0.05). Statistical analysis was carried out using the IBM SPSS statistics version 23.0 program (IBM Corp., 2021, Armond, NY, USA).

## 3. Results

The parameters, including the mean error, the standard deviation, and the minimum and maximum values, of each of the 28 distances studied for each suprastructure are shown for the maxillary in Table 1 and for the mandibular in Table 2.

When analyzing the precision data according to the selected arbitrary limits (75 μm for the distances and 0.6 degrees for the relative angulation of the abutments) the following results organized from best to worst were found: The percentage of distances that were below the 75 μm arbitrary limit were for maxillary and mandibular, respectively: 98.6% and 95.4% for PIC, 90.4% and 80.0% for TD, and 77.9% and 68.2% for T. The percentage of angulations below the arbitrary limit of 0.6 degrees were; 100% and 100% for PIC, 91.4% and 81.4% for TD, and 87.1% and 55.7% for T (Table 3).

For the TD and T systems, when the distance between the abutments increased, the error of each parameter studied increased as well, being more evident between the Euclidean distances from pillar 1 to 8 (Table 4 and Figure 4, Figure 5 and Figure 6). In the TD system, a similar pattern was observed for both arches; however, T performed worse in the mandibular arch than in the maxillary arch. The IOS PIC was not affected by the increase in the distance between the abutments. The greatest difference between measurements for a segment in the maxillary arch (Table 1) was PIC 108.2 μm, T 220.6 μm, and TD 342.9 μm and for the mandibular arch (Table 2) PIC 104.4 μm, TD 269.0 μm, and T 476.4 μm.

Table 5 and Table 6 show the same parameters for the relative angulations of each abutment in the maxillary and mandibular bars, respectively. The PIC system was also not affected by the increase in the distance between implants. The TD system showed a certain increase in the mean error and DS in both arches as the separation between the abutments increased. The T system did not present an incremental pattern; however, the mandible presented 31.4% more data over 0.6 degrees than the maxillary (Figure 7 and Figure 8).

The greatest difference between measurements for an abutment in the maxillary arch (Table 5) was 0.44 degrees for PIC, 1.01 degrees for TD, and 1.26 degrees for T; in the mandibular arch (Table 6), it was 0.24 degrees for PIC, 1.61 degrees for TD, and 2.57 degrees for T.

The distances of the 28 segments were compared between the different IOSs (PIC, T, and TD) Significant differences were found in Euclidean distances when comparing PIC–T in 14 segments for the maxillary arch and 11 for the mandibular arch. When comparing PIC–TD, 14 segments were different in the maxillary and 11 in the mandibular. Comparing TD–T, 11 segments were different in the maxillary and in the mandibular (Table 7).

Comparing the angulations, we observed significant differences when comparing PIC–T in four abutments of the maxillary arch and in six of the mandibular arch. When comparing PIC–TD, there were in two maxillary and six mandibular, and when comparing TD–T, there were in two maxillary and one mandibular abutment (Table 8).

## 4. Discussion

Based on the results of the present study, the three studied IOSs showed different precision when compared with the pre-established limits. Therefore, the first hypothesis was rejected. The precision of the systems T and TD decreased when the distance between the implants increased; this is consistent with multiple in vitro studies [3,21,39,55]. This was more evident for the distances than for the angles; however, the PIC system was not affected by the increasing distance. Therefore, the second hypothesis was partially rejected. The T and TD systems again showed different results depending on the arch, obtaining worse precision in the mandibular arch than in the maxillary. The PIC obtained similar results for both arches. Therefore, the third hypothesis was partially rejected.

Since this is an in vivo study, the accuracy cannot be evaluated because it is not possible to obtain a gold standard; however, it is possible to analyze the precision, and it provides useful information regarding the predictability and degree of variability of the different digital impressions. The purpose of analyzing the trueness and precision of the impression is to know whether the IOS can obtain a digital working cast to manufacture prosthetic structures with passive adjustment. This was completed without taking into account the possible errors of the rest of the CAD–CAM digital workflow [56].

One interesting debate continues about the tolerable range of acceptable misfit for our suprastructures. The literature is once again heterogeneous, both about how to assess passive fit as well as the tolerable ranges of the misfit. It would be very beneficial to specify the limit of misfit and, above all, under what parameters [53]. These limits can be established according to the three-dimensional variation of each implant, the distance between them, the individual and relative angulation, etc. [1,6,31,33]. Manzella et al. studied the effect of the three-dimensional and angular variation of a single implant in a structure with four and six implants. They established limits of 150 μm in the horizontal plane, 50 μm in the vertical, and 1 degree in the variation of the angulation [53]. More studies of this kind, in which the movements of all the implants involved in a suprastructure are combined, would be necessary, since it would be logical to think that the greater the number of implants, the lower the misfit tolerance for each segment. The acceptable misfit in the literature varies from 50 to 150 μm. The studies use different methodologies to measure the deviations, some report the data in the 3 axis (X, Y, Z), others with one single point evaluating distances and angles, and another very different methodology uses the best fit algorithm and compares the scanned Scanbodies. At the moment, there is no consensus of a clinically acceptable tolerance range [13,43]. In the present study, in order to help the reader interpret the results, the authors decided to select an arbitrary limit, based on the acceptable misfit published in previous articles. As the vast majority of studies are carried out on four or six implants, and the present was eight, stricter limits were selected by reducing by half those described by Manzella [53]. The limit between two implants was placed at 75 μm for the distance and at 0.6 degrees for the relative angulation. In this way, the reader can see that if there was an established limit, the amount of times that the fit would not reach the limit, as a reference without affirming whether or not it has clinical relevance, since there is no such limit established in the literature yet.

In a multiple implant prosthesis, it is essential to study the Euclidean distances together with the angulations, since there are infinite angular positions of the abutments that can affect the fit of the structure even when they have the same distances [6,41]. In addition, we must analyze the precision/trueness, not only based on the data obtained by analyzing the mean deviation of the variables to study (distances or angulations), since these measurements compensate for one another, and the resulting data can hide or greatly minimize the errors, leading to situations in which almost no deviation is found, but when analyzing each variable individually, there are significantly more discrepancies [23] (Figure 9). As an example, when comparing the arithmetic means of the 28 Euclidean distances of the first and fifth impressions for the mandibular arch with the T system, a difference of 103 μm was obtained. However, when comparing each section individually, a difference of 167 μm was observed between Section 1 and Section 5, 246 μm between 1 and 6, 317 μm between 1 and 7, and 477 μm between 1 and 8.

The comparison of 3D files using mesh alignment techniques (Best Fit) is used in many studies, since the process is relatively simple and gives a visual qualitative/quantitative idea of the discrepancies [7,8]. This process compensates for the discrepancies when the meshes align, as these do not share either the number, the size, or the distribution or form of triangles [21,30,32,33,57]. In addition, the results may be affected by the Best Fit algorithm used by each software, the alignment method, and above all by the amount of STL mesh that is selected in the process [31,32,33]. In the present study, the center points and rotational axes of the abutments of the suprastructures were calculated to analyze the precision of each scanner. The results would be the same as if performed on the measurements of the virtual replicas of the implants, since there is no virtual space between them; in this way, we have reached the last step of the CAD sequence. Every study contributes and increases knowledge, and it is subject to variations in its results due to errors and/or differences in the instruments and software used, hence the need to standardize them and craft protocol [21].

According to van der Meer, all the intraoral scanners, including TD and T build their 3D models by combining several 3D images made of the same section of the model but from different angles. [14]. When the field of view decreases in size, the scan area increases, and more images are needed. Therefore, the scan protocol used and the starting point of the scan affect the precision of the impression. The images are stitched together by the algorithms that manage the scanner, increasing the definitive error when performing the 3D recomposition [21,31,42]. That is why our results agree with in vitro studies such as those of Gimenez, where this effect was already described [39,41,42,49]. As the scanner moves away from the starting point, there is an increasing pattern of imprecision. The standard error and the SD increase and, in particular, the maximum error between measurements. Another process that these two systems have to undergo is the transformation of the STL file to the digital working cast through the “Best Fit” alignment with the implant library in CAD design software. The effect of this step is proportional to the number of scan bodies that must be aligned. The PIC system lacks both steps, so when analyzing the data, it was observed that the error did not increase as we moved away from the starting point, in the way that happened in the other systems studied [42] (Figure 5). It is important to differentiate photogrammetry technology, in general, from the PIC system; they share similar physical principles, but in the PIC system, there is software linked to the cameras carrying specific algorithms adjusted to the oral environment [44,45,46,47,48].

The effect of the separation between implants and the consequent presence of edentulous areas are also factors, already described in the literature, that affect IOSs such as T and TD [9,37,42]. The lack of anatomical references for the soft tissues, added to the fact that the scan bodies used usually have the same exact design, causes the systems to lose the position when sewing the photos due to the lack of specific recognition areas [2,35]. The maxillary arch that has more keratinized tissue, along with the palate and its rugs can be factors that could help improve the impression precision [27,35]. In the present study, the authors think that another factor that could influence poorer results in the mandibular arch was the position of the eight implants. They presented less distance between them than in the maxillary arch; thus, it was challenging to scan interproximal areas of the scan bodies [2]. The PIC system was not affected by the arch type either, as shown in the results. As long as the two lenses can visualize the reference abutments and the rest of the PIC Transfers^®^, the registration will be completed regardless of the soft tissue. In our case, it was achieved through small movements of the PIC camera^®^, even though it is also possible to move the patient or rotate the PIC transfers (scan bodies) as needed.

The results, using the arbitrary limits selected, showed that the PIC system especially indicated for multiple implants cases had the best precision [43]. Another factor benefitting this system is that it does not depend on any scan pattern, therefore, being less sensitive to the operator [28]. On the other hand, TD performed second in precision but was the one more clearly affected by the increasing distance and had the disadvantage of using powder, even though as Kim et al. stated, systems that use powder are more accurate at happened in this study [21]. The T system was the one that obtained the worst precision data both in distances and in angulations, where it was more evident. It was observed by the authors, that the second impression of the mandibular arch was responsible for making the results worse. The operator had difficulties completing the impression following the scan pattern recommended by the manufacturer. This was not ruled out to avoid introducing any bias. On the other hand, it should be noted that T never offered its system for this type of treatment, specifying in its indications bridges on implants with a maximum of three pieces, where in different studies it obtained good results [21]. Despite the limitations of IOS for taking implant impressions, there are several clinical publications using IOS for a full arch already [58,59,60,61,62,63,64,65], and presentations showing these clinical cases performed with optical impressions are being regularly shown in dental congresses. That was one of the main reasons for comparing PIC with IOS in this in vivo study.

There are other variables that can influence the optical impression systems results; the type of implant connection being one of them, as well as the level at which the impression is taken (to an intermediate abutment or directly to the implant) as described previously in the dental literature, for conventional [66,67] as well as for optical impressions. Two systematic reviews of optical impressions mention these factors. Rütkunas et al. considered that these factors influenced the accuracy of optical impressions [2]. Zhang concluded that the type of connection and depth of the implants did not affect the accuracy [65].

The effect of the room light on IOS has been studied recently [68,69,70]. Studies show that it affected both mesh quality and accuracy, so room light must be adapted to each intraoral scanner. In the present study, the setting was a regular clinical room, and we followed the recommendations and instructions of the manufacturer that did not include any specifics about room light (just not to use the chair light).

A layer of titanium dioxide powder was needed in the True Definition IOS used in this study. The thickness of this layer has been calculated to be 20 to 40 microns and avoids light reflection [27]. Some authors, such as Nedelcu [71], state that this layer did not affect the final result of the impression. However, Rutkūnas [2] considered it to be a conditioning factor. Kim [21], in his study of the accuracy of nine intraoral scanners for complete-arch image acquisition, considered the IOS using this titanium dioxide powder the most accurate one, which was consistent with the results of this study.

As can be seen, there were significant differences (*p* < 0.05) between the Euclidean distances and angulations when comparing the scanners with each other [8,11]. This indicates that there is no reproducibility; that is, there is no precision between the IOS studied [8,11]. More in vivo studies are needed to evaluate whether optical printing systems can replace conventional impressions for this type of treatment. We propose studies with another number and distribution of implants, the use of other implant brands, as well as the use of intermediate abutments, different Scanbodies, different room lights, and IOSs with more developed software. For all this, the results of this study must be interpreted with caution.

## 5. Conclusions

Within the limitations of the present in vivo study, the following conclusions are drawn:The system that obtained better precision was PIC, followed by TD, and then T.The precision of T and TD decreased as the distance between the implants increased; however, this variable did not affect the PIC system.The arch did not affect PIC precision, but the T and TD systems performed worse in the mandibular arch.

## Figures and Tables

**Figure 1 ijerph-19-04300-f001:**
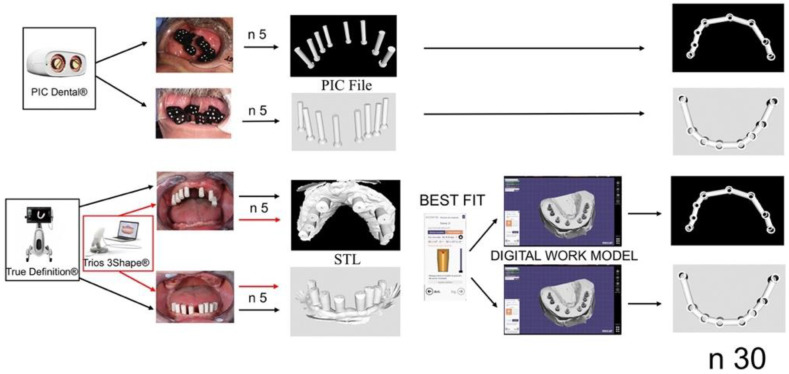
Diagram representing the study design. Three IOSs systems, 5 scans per arch, maxillary mandibular, digital study model, and suprastructure design.

**Figure 2 ijerph-19-04300-f002:**
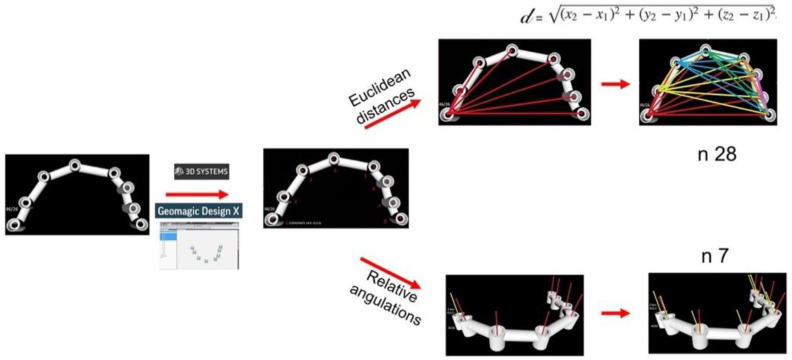
With reverse engineering software, center points and rotational axis of abutments are calculated for 30 designed suprastructures. From each suprastructure 28 Euclidean distances and 7 relative angulations were analyzed.

**Figure 3 ijerph-19-04300-f003:**
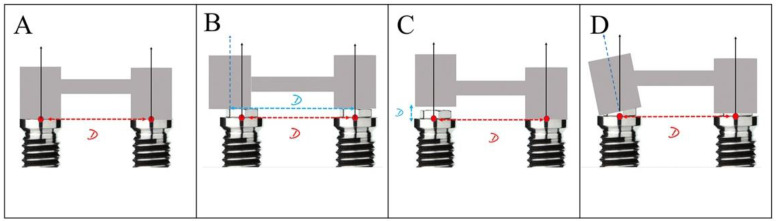
Representation of fit of suprastructure over two implants. (**A**) Ideal passive fit. (**B**) Misfit due to deviation on X axis between abutments. (**C**) Misfit due to deviation on Y axis between abutments. (**D**) Misfit due to variation of angulation between abutments, depending on the number of abutments involved in each prosthetic structure.

**Figure 4 ijerph-19-04300-f004:**
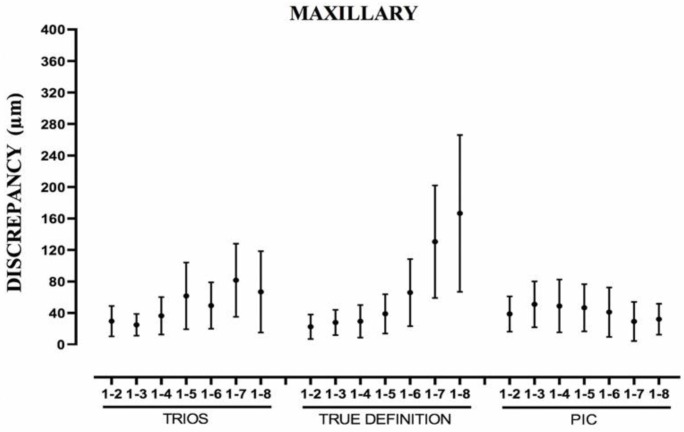
Boxplot showing measurements of Euclidean distances from distance 1-2 to 1-8 of maxillary arch. Point represents mean error; line represents standard deviation.

**Figure 5 ijerph-19-04300-f005:**
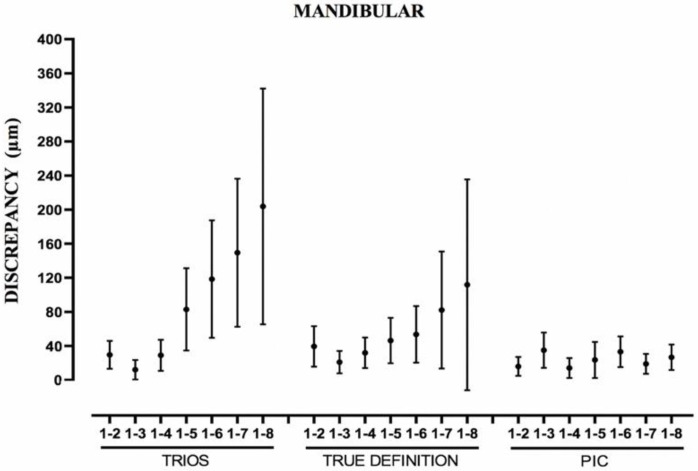
Boxplot showing measurements of Euclidean distances from distance 1-2 to 1-8 of mandibular arch. Point represents mean error; line represents standard deviation.

**Figure 6 ijerph-19-04300-f006:**
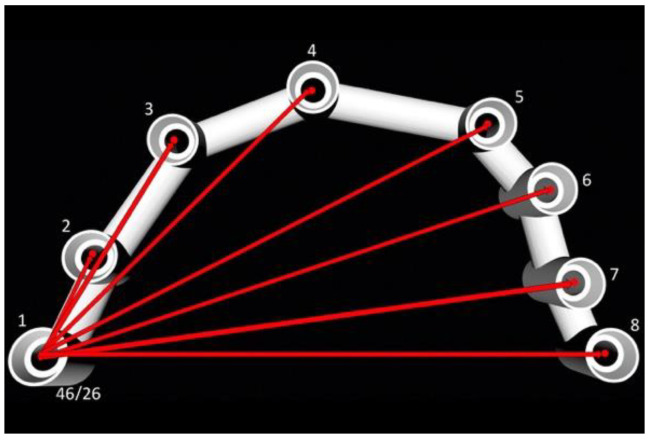
Representation of the Euclidean distances from abutments 1 to 8.

**Figure 7 ijerph-19-04300-f007:**
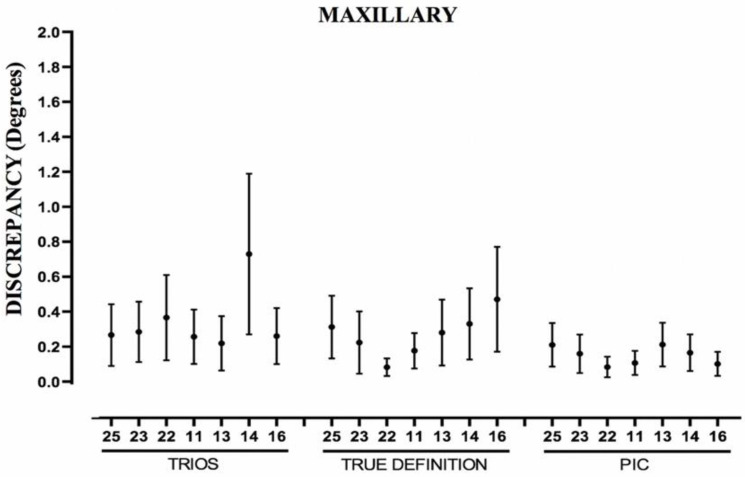
Boxplot showing measurements of relative angulations of abutments of maxillary arch. Abutment 26 is used as reference. Point represents mean error; line represents standard deviation.

**Figure 8 ijerph-19-04300-f008:**
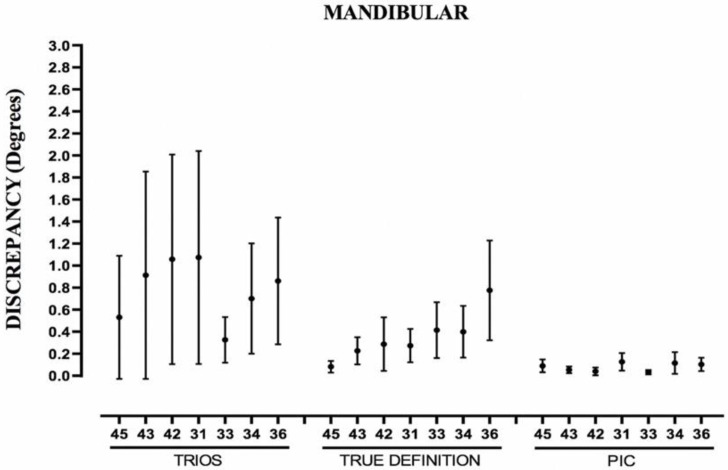
Boxplot showing measurements of relative angulations of abutments of mandibular arch. Abutment 46 is used as reference. Point represents mean error; line represents standard deviation.

**Figure 9 ijerph-19-04300-f009:**
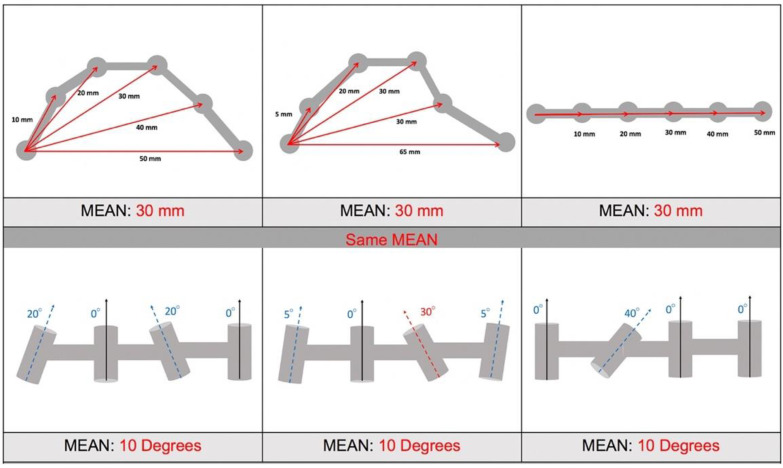
Analysis of trueness/precision calculated with each distance individually is more representative of real distortions than the means. See in the image that the same mean error can mean very different level of individual deviations, with very different clinical implications.

**Table 1 ijerph-19-04300-t001:** Descriptive statistics for each of 28 Euclidean distances (μm) of maxillary arch obtained by the three evaluated systems.

Euclidean Distances	PIC Dental				TRIOS				True Definition			
	Mean Error	DS	Min	Max	Mean Error	DS	Min	Max	Mean Error	DS	Min	Max
**1-2**	38.66	22.40	10.09	74.39	38.66	22.40	10.09	74.39	22.55	15.56	1.11	38.72
**1-3**	51.00	29.15	12.20	99.64	24.96	13.70	6.88	51.57	27.91	16.00	3.11	50.97
**1-4**	48.91	33.63	9.96	103.11	36.74	23.76	8.23	76.21	29.36	20.63	1.44	58.26
**1-5**	46.61	29.91	8.17	107.68	61.68	42.37	5.46	144.90	38.85	24.90	4.72	68.64
**1-6**	41.01	31.38	1.39	96.99	49.37	29.42	7.50	94.53	65.86	42.58	9.04	139.20
**1-7**	29.16	24.86	4.52	66.98	81.58	46.48	11.07	157.96	130.45	71.44	32.47	264.42
**1-8**	32.03	19.63	4.02	64.59	66.79	51.68	1.75	156.86	166.46	99.60	18.19	342.97
**2-3**	30.36	22.68	2.68	62.16	46.06	30.83	1.00	79.75	7.59	4.61	0.06	7.64
**2-4**	33.57	21.02	5.93	60.66	60.55	40.37	6.26	139.88	31.62	16.80	12.59	61.52
**2-5**	18.71	10.92	0.92	38.34	92.26	56.15	28.09	196.17	35.33	22.33	1.48	70.26
**2-6**	30.47	20.66	3.13	55.91	58.59	38.51	4.77	134.01	43.91	40.78	2.85	104.28
**2-7**	26.63	16.52	1.25	51.75	92.09	53.67	5.75	174.30	102.06	65.73	3.75	234.75
**2-8**	23.56	16.54	2.16	53.79	61.35	49.19	4.99	138.72	137.89	88.67	15.17	310.05
**3-4**	30.79	18.65	9.47	64.31	39.45	22.31	8.64	79.50	29.33	17.62	6.49	59.88
**3-5**	29.67	23.21	1.35	64.08	75.82	46.83	14.01	152.16	17.85	11.41	0.36	37.26
**3-6**	20.57	12.55	4.08	42.30	55.39	34.05	4.33	108.86	183.60	14.97	2.87	41.14
**3-7**	11.73	7.66	0.96	24.03	113.98	63.95	19.53	220.65	64.66	41.65	9.56	142.33
**3-8**	20.34	12.60	0.78	35.59	69.45	53.29	11.21	152.02	90.89	68.66	6.08	204.10
**4-5**	27.77	18.78	3.93	51.83	83.99	49.65	11.95	174.90	37.40	23.68	0.74	73.38
**4-6**	30.09	20.03	2.74	70.06	83.35	50.80	6.04	150.46	35.78	21.94	12.07	76.11
**4-7**	25.13	22.66	1.57	55.60	120.00	85.52	1.48	205.40	20.87	14.14	1.89	43.97
**4-8**	44.00	27.74	6.11	88.36	32.73	21.39	5.96	71.14	28.50	18.29	0.03	63.90
**5-6**	39.68	29.49	2.25	71.89	32.94	21.18	5.57	64.65	39.69	38.07	1.98	96.83
**5-7**	24.96	14.53	4.64	54.94	69.03	38.51	13.09	128.26	44.27	38.81	0.92	106.74
**5-8**	33.62	23.11	4.37	77.31	45.72	32.25	6.44	98.08	36.86	24.38	6.46	85.65
**6-7**	23.46	14.05	2.78	50.71	68.95	41.06	11.55	125.95	145.19	8.84	2.06	29.40
**6-8**	50.77	31.85	16.86	108.26	48.24	32.41	0.45	89.80	11.86	6.84	2.97	22.63
**7-8**	26.74	18.42	1.41	56.51	101.62	60.48	21.32	193.56	18.74	11.61	0.04	41.98

Standard Deviation (DS); Minimum (Min); Maximum (Max).

**Table 2 ijerph-19-04300-t002:** Descriptive statistics for each of 28 Euclidean distances (μm) of mandibular arch obtained by the three systems evaluated.

Euclidean Distances	PIC Dental				TRIOS				True Definition			
	Mean Error	DS	Min	Max	Mean Error	DS	Min	Max	Mean Error	DS	Min	Max
**1-2**	15.98	11.06	1.27	29.71	29.53	16.38	7.05	61.41	39.50	23.74	4.58	82.39
**1-3**	34.98	20.66	8.34	66.59	12.06	11.42	0.62	27.14	20.91	13.12	1.16	40.58
**1-4**	14.10	11.63	2.73	31.74	29.01	18.12	4.14	52.25	31.99	17.96	5.31	64.68
**1-5**	23.63	21.14	2.15	55.70	82.89	48.36	6.95	166.68	46.27	26.65	15.93	98.79
**1-6**	33.24	18.00	11.54	66.76	118.42	68.95	6.94	246.55	53.58	33.25	7.70	118.95
**1-7**	18.96	11.62	0.04	36.33	149.41	86.95	40.46	316.54	82.16	68.64	1.84	18.63
**1-8**	26.74	14.91	9.04	53.20	203.70	138.40	20.51	476.46	111.76	123.75	3.50	269.08
**2-3**	44.45	26.88	7.20	98.68	19.32	10.91	5.28	40.92	10.70	7.87	1.29	24.91
**2-4**	34.40	19.40	4.77	64.78	28.85	26.24	0.22	64.92	13.70	8.77	2.02	26.71
**2-5**	35.43	20.47	7.80	66.01	45.41	27.42	0.57	80.32	19.22	11.05	1.76	40.94
**2-6**	46.56	30.68	1.46	104.43	62.60	36.29	5.60	117.66	27.52	16.94	6.00	58.96
**2-7**	41.31	30.58	4.81	89.50	75.76	45.95	13.00	154.74	37.50	22.31	9.21	69.78
**2-8**	33.59	24.81	1.63	70.92	118.31	78.34	6.12	272.59	61.95	35.90	13.09	113.61
**3-4**	29.37	18.53	4.39	53.76	16.45	12.81	0.31	32.04	11.91	7.34	0.85	23.44
**3-5**	26.64	16.64	0.78	46.29	32.71	18.24	8.18	64.78	15.15	10.05	1.19	34.03
**3-6**	31.79	19.51	0.69	70.99	41.32	27.42	0.38	85.19	27.13	14.98	6.29	57.39
**3-7**	16.35	11.26	1.28	38.18	50.19	31.22	13.95	104.19	36.90	21.05	5.78	67.82
**3-8**	12.29	9.76	0.69	27.76	93.91	55.71	18.11	202.69	59.43	33.09	11.54	115.61
**4-5**	24.76	17.72	2.53	56.63	19.69	11.49	3.73	38.22	15.21	18.25	0.21	37.79
**4-6**	33.46	24.36	2.54	64.97	24.85	16.96	0.04	50.67	24.62	17.71	1.41	56.66
**4-7**	23.47	15.82	0.26	51.82	34.49	18.91	11.32	66.25	27.84	16.60	2.37	54.90
**4-8**	8.88	7.27	0.59	20.64	70.98	41.74	12.17	149.44	43.58	24.41	11.61	86.01
**5-6**	15.79	12.22	2.77	35.79	8.37	6.09	0.44	20.29	14.48	9.95	1.13	34.13
**5-7**	16.68	10.10	0.20	36.40	13.52	7.48	2.65	25.55	19.11	11.77	2.05	43.23
**5-8**	28.67	22.08	0.63	53.81	54.79	32.93	12.74	120.29	42.24	26.37	0.40	82.73
**6-7**	27.36	17.76	0.82	51.37	7.75	6.39	0.25	17.07	6.06	3.74	1.08	11.47
**6-8**	27.34	22.44	3.86	61.37	42.87	31.96	3.73	102.98	29.88	22.46	1.16	67.12
**7-8**	21.31	13.25	4.76	44.49	47.89	30.94	3.86	110.73	41.13	33.35	0.00	31.17

Standard Deviation (DS); Minimum (Min); Maximum (Max).

**Table 3 ijerph-19-04300-t003:** Descriptive statistics where number of distances and angulations that were above/below the previously established limits are shown.

		Distances (µm)			Angulations (Degrees)		
		>75	<75	% < 75	>0.6	<0.6	% < 0.6
TRIOS	Maxillary	62	218	77.9	9	61	87.1
	Mandibular	89	191	68.2	31	39	55.7
True Definition	Maxillary	27	253	90.4	6	64	91.4
	Mandibular	56	224	80.0	13	57	81.4
PIC Dental	Maxillary	4	276	98.6	0	70	100
	Mandibular	13	267	95.4	0	70	100

**Table 4 ijerph-19-04300-t004:** Maximum errors of distance measurements of each IOS in each arch. Distance 1-8 was selected to represent the distance that was most affected by accumulative error.

Euclidean Distance	1-2	1-3	1-4	1-5	1-6	1-7	1-8
TRIOS	Maxillary	74.4	51.7	76.2	144.9	94.5	157.9	156.8
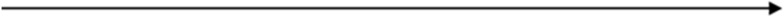
Mandibular	61.4	27.1	52.2	166.6	246.5	316.5	476.4
True Definition	Maxillary	38.7	50.9	58.2	68.6	139.2	264.4	342.9 +
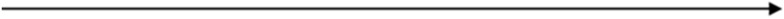
Mandibular	82.3	40.5	64.6	98.7	118.9	18.6	269.0
PIC Dental	Maxillary	74.3	99.6	103.1	107.6	96.9	66.9	64.5
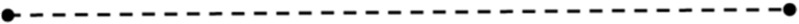
Mandibular	29.7	66.5	31.7	55.7	66.7	36.3	53.2

**Table 5 ijerph-19-04300-t005:** Descriptive statistics for each of seven relative angulations (degrees) of maxillary arch obtained with the three evaluated systems.

Angulation Deviations	PIC Dental				TRIOS				Angulation Deviations			
	Mean Errror	DS	Min	Max	Mean Errror	DS	Min	Max	Mean Errror	DS	Min	Max
**26**	0	0	0	0	0	0	0	0	0	0	0	0
**25**	0.21	0.12	0.01	0.41	0.26	0.17	0.01	0.55	0.31	0.17	0.04	0.64
**23**	0.15	0.11	0.01	0.35	0.28	0.17	0.01	0.52	0.22	0.17	0.00	0.52
**22**	0.08	0.05	0.01	0.17	0.36	0.24	0.04	0.74	0.08	0.05	0.01	0.17
**11**	0.10	0.06	0.00	0.24	0.25	0.15	0.05	0.57	0.17	0.10	0.01	0.36
**13**	0.21	0.12	0.03	0.44	0.21	0.15	0.02	0.52	0.28	0.18	0.05	0.60
**14**	0.16	0.10	0.02	0.34	0.72	0.45	0.00	1.26	0.32	0.20	0.03	0.70
**16**	0.10	0.06	0.00	0.20	0.26	0.16	0.04	0.46	0.47	0.29	0.07	1.01

Standard Deviation (DS); Minimum (Min); Maximum (Max).

**Table 6 ijerph-19-04300-t006:** Descriptive statistics for each of seven relative angulations (degrees) of mandibular arch obtained with the three evaluated systems.

Angulation Deviations	PIC Dental				TRIOS				True Definition			
	Mean Error	DS	Min	Max	Mean Error	DS	Min	Max	Mean Error	DS	Min	Max
**46**	0	0	0	0	0	0	0	0	0	0	0	0
**45**	0.08	0.05	0.00	0.20	0.53	0.55	0.02	1.25	0.08	0.05	0.01	0.17
**43**	0.05	0.03	0.01	0.11	0.91	0.94	0.03	2.21	0.22	0.12	0.06	0.43
**42**	0.04	0.03	0.00	0.09	1.05	0.95	0.05	2.49	0.28	0.24	0.00	0.62
**31**	0.12	0.07	0.00	0.21	1.07	0.96	0.04	2.57	0.27	0.15	0.07	0.56
**33**	0.03	0.01	0.00	0.06	0.32	0.20	0.07	0.73	0.41	0.25	0.04	0.72
**34**	0.11	0.09	0.00	0.24	0.70	0.50	0.06	1.64	0.39	0.23	0.05	0.87
**36**	0.10	0.60	0.02	0.19	0.86	0.57	0.12	1.99	0.77	0.45	0.24	1.61

Standard Deviation (DS); Minimum (Min); Maximum (Max).

**Table 7 ijerph-19-04300-t007:** *p*-values of the comparisons of distance measurements between three IOSs in maxillary/mandibular arch. Significant *p*-values have been highlighted with bold numbers.

Euclidean Distances	PIC vs. TRIOS		PIC vs. Euclidean Distances		True Definition vs. TRIOS	
	Maxillary *p*-Value	Mandibular *p*-Value	Maxillary *p*-Value	Mandibular *p*-Value	Maxillary *p*-Value	Mandibular *p*-Value
**1-2**	0.528	0.454	0.146	0.210	0.384	0.491
**1-3**	**0.013**	**0.022**	**0.034**	0.057	0.585	0.859
**1-4**	0.252	0.060	0.117	0.245	0.608	0.733
**1-5**	0.327	0.073	0.975	0.733	0.263	0.137
**1-6**	0.662	**0.012**	0.528	0.092	0.354	0.087
**1-7**	0.162	**0.002**	**0.008**	**<0.001**	0.140	0.760
**1-8**	**0.002**	**0.003**	**<0.001**	**<0.001**	**0.035**	0.723
**2-3**	0.099	**0.030**	**0.004**	**0.011**	**<0.001**	0.469
**2-4**	0.153	0.063	0.219	**0.039**	0.052	**<0.001**
**2-5**	**0.002**	0.358	**0.031**	**0.016**	**0.018**	**0.010**
**2-6**	0.178	0.685	**<0.001**	0.081	0.343	0.062
**2-7**	**0.003**	0.544	**0.026**	0.203	0.839	0.197
**2-8**	**<0.001**	**0.018**	**0.002**	0.129	0.118	0.072
**3-4**	0.866	0.079	0.804	**0.001**	0.716	0.119
**3-5**	**0.034**	0.934	**0.012**	0.102	**0.001**	0.139
**3-6**	0.100	0.551	0.209	0.596	0.214	0.307
**3-7**	**<0.001**	**0.009**	**0.001**	**0.047**	0.174	0.220
**3-8**	**<0.001**	**0.001**	**<0.001**	**0.006**	0.282	0.104
**4-5**	**0.006**	0.132	0.873	0.457	**0.013**	**0.002**
**4-6**	**0.007**	0.237	0.655	0.313	**0.012**	**0.811**
**4-7**	**<0.001**	0.406	0.020	0.847	**<0.001**	**0.526**
**4-8**	0.345	**<0.001**	0.135	**0.002**	0.526	**0.075**
**5-6**	0.054	**0.004**	0.102	0.194	**0.003**	**0.215**
**5-7**	**0.007**	0.606	**<0.001**	0.670	0.712	**0.337**
**5-8**	0.216	0.313	0.940	0.808	0.232	**0.484**
**6-7**	**0.001**	**0.008**	0.094	**0.001**	**<0.001**	**0.018**
**6-8**	0.752	0.481	**0.002**	0.837	**<0.001**	**0.427**
**7-8**	**0.001**	0.102	0.190	**0.002**	**<0.001**	**0.408**

**Table 8 ijerph-19-04300-t008:** *p* values of the comparisons of each of seven relative angulations for both arches with three different IOSs combination. Significant *p*-values highlighted with bold numbers.

Abutment Maxillary/Mandibular	PIC vs. TRIOS		PIC vs. Maxillary/Mandibular		True Definition vs. TRIOS	
	Maxillary *p*-Value	Mandibular *p*-Value	Maxillary *p*-Value	Mandibular *p*-Value	Maxillary *p*-Value	Mandibular *p*-Value
**26/46**	-	-	-	-	-	-
**25/45**	0.425	0.070	0.158	0.772	0.571	0.070
**23/43**	0.070	**0.002**	0.348	**<0.001**	0.447	0.257
**22/42**	**0.002**	**0.001**	0.973	**0.005**	**0.002**	0.059
**11/31**	**0.012**	**0.004**	0.088	**0.014**	0.190	**0.049**
**13/33**	0.909	**<0.001**	0.350	**<0.001**	0.438	0.405
**14/34**	**0.001**	**0.002**	**0.035**	**0.002**	**0.022**	0.102
**16/36**	**0.010**	**0.001**	**0.001**	**<0.001**	0.066	0.716

## Data Availability

The data presented in this study are available on request from the corresponding author. The data are not publicly available due to privacy.

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
