# Peer review of "In Vivo Complete-Arch Implant Digital Impressions: Comparison of the Precision of Three Optical Impression Systems"

_ijerph, 2022, doi:10.3390/ijerph19074300_

Round 1

Reviewer 1 Report

  1. The main remark concerns the aim of the study – is it correct to compare the technique specially designed for digital determination of the implants position in the edentulous jaw (PIC) with the techniques that are not intended for these purposes. The result of such a study is predetermined in advance. It would be more logical to compare the results of using the PIC scanner with the traditional analog technique for determining the position of implants. Have such studies been conducted before? If so, it would be appropriate to point them out in the discussion.
  2. Requires clarification of the authors' statement that “TD and T systems base their technology on the acquisition of 2D images and create from them a 3D file”(328)

Reviewer 2 Report

Authors can improve the presentation of results. Trying to clarify and justify them, if possible discussing better with the existing literature.

Reviewer 3 Report

This submitted draft would seem easily intelligible, and is considered worth following after a thorough revision

Abstract

Please add the exact p value

Intro

- Objectives would seem well-elaborated.

Materials and Methods

- Did the study approve by ethical committee? Please clearly clarify about it as the ethical approval in text related with the study was in 2014, should be a update?

- Please provide MPI (brand name; manufacturer, city, country)

- Figure caption must follow journal guideline, must be double checked.

- The author should explain the sample size calculation?

Results

- Where appropriate, please add exact p values with your text.

- Again, please clearly clarify the stat significance with exact p value in the table 1 to 6 and figure 4, 7 and 8 related to the stat analysis

- Table 7 should be revised

Discussion

- In the present study, the authors used 8 external hex implants (multiple implant prothesis with ext hex), so it should be better to discuss about other connection design implants with accuracy of IOCs?

Conclusion

- This section must not be overgeneralized and should be based on the present study result.

References

- must follow the journal guidelines (do not shorten the page number)

It is strongly recommended that the revised manuscript is reviewed to correct any grammatical or spelling errors.

Reviewer 4 Report

It is recommended to send the study to the Journal of Functional Biomaterials since the subject would contribute to the community of the readers in that field.

The manuscript is clear and well-structured of little relevance in the field of environment and public health, of moderate relevance in the field of implantology, on the one hand there is little evidence of the tolerable limits of misalignment, however, each system should have its own limits inherent to its mechanism and these should not be generalized unless apply the same digital printing technology.

Arbitrary limits should be set for IOSs for full arches and for those that only scan isolated implants.

The arbitrary limits selected are questionable when they were only based on one publication (Manzella 2016) to determine this precision and of a smaller segment that goes from 4 to 6 implants and not 8 as in this case.

It is suggested to cite in the text the articles on which they were based. It would be advisable to present a table with these results since this data is a reference of its possible results.

What was the selection of this arbitrary limit of acceptable misfit based on? Was a systematic review carried out?

One of the hypotheses mentions that the precision is not affected by the type of arch and in the abstract, it is mentioned within the objectives of the study that the influence of the factors in the maxillary and mandibular arch will be determined, which casts doubt on the congruence of this hypothesis.

The factors that influenced each arch, the discrepancy of the results and the decrease in performance in the mandibular arch are not mentioned.

What type of light was used when performing the scan? How was it determined how much this might influence the accuracy? It is suggested to mention in the materials and methods section how they controlled this factor.

It is suggest to mention how the length of the edentulous section influenced each scanner used, especially in the T and TD that are not specific for these cases and if it had an impact on the stitching of the 2D images to produce the STL file?

Why was the thickness of he film created by the titanium dioxide powder not taken into account and how does it influence the results of the arbitrary limits? Mention it in the discussion.

There is a general scope that force the inference of the expecting results, thus possible bias to the reader can be expected to avoid this, the authors should try to keep an objective writing style in the first part of the paper.
